# Treatment outside the Recommended Guidelines for Retinopathy of Prematurity (ROP): Prevalence, Characteristics, and Issues

**DOI:** 10.3390/jcm11010039

**Published:** 2021-12-22

**Authors:** Daniel Lemaître, Amandine Barjol, Youssef Abdelmassih, Caroline Farnoux, Gilles C. Martin, Florence Metge, Thibaut Chapron, Georges Caputo

**Affiliations:** 1Pediatric Ophthalmology Department, Rothschild Foundation Hospital, 75019 Paris, France; dlemaitre@for.paris (D.L.); abarjol@for.paris (A.B.); yabdelmassih@for.paris (Y.A.); gmartin@for.paris (G.C.M.); fmetge@for.paris (F.M.); gcaputo@for.paris (G.C.); 2Faculty of Medicine, Sorbonne Université, 75006 Paris, France; 3Department of Pediatric Surgery, AP-HP, Hôpital Robert Debre, 75019 Paris, France; caroline.farnoux@aphp.fr; 4Epidemiology and Statistics Research Center/CRESS, INSERM, INRA, University of Paris, 75004 Paris, France

**Keywords:** retinopathy of prematurity, laser photoablation, bevacizumab IVI, guidelines, ETROP

## Abstract

This study aims to assess the prevalence and characteristics of preterm infants with retinopathy of prematurity (ROP) treated outside the recommended guidelines. In this retrospective monocentric cohort, we included all premature children treated in our department for ROP by laser photoablation or anti-VEGF intravitreal injection. The main outcome was treatment of both eyes for ROP less severe than pre-threshold type 1, treated outside ETROP guidelines. A total of 114 children received treatment for ROP in our department, among whom 32 (28.1%) children received treatment for indications outside the ETROP guidelines for both eyes. The indications outside the guidelines were persistent stage 2 or 3 ROP that showed no evidence of regression after 41 weeks of corrected gestational age (11 children; 34.4%), pre-plus stage (11; 34.4%), difficulties in disease staging (7; 21.9%), type 2 ROP with plus disease (2; 6.2%), and treatment due to logistical difficulties (1; 3.1%; hospitalized in neonatal units hundreds of miles away from our department, with no fundus examination possible in the neonatal unit). To resume, in our cohort, 28.1% of children received treatment for ROP less severe than pre-threshold type 1 both eyes. The main indications for off-label treatment were the persistence of active ROP during follow-up and the presence of pre-plus-stage disease. Our data suggest the need to update ROP treatment criteria to reflect real-life practices. Additional studies are required in order to evaluate the long-term benefits and side effects of treatments outside the recommended indications, and to establish revised treatment guidelines.

## 1. Introduction

Over the past decade, the survival of preterm children has improved; however, these children remain vulnerable to neonatal complications—especially retinopathy of prematurity (ROP) [1,2,3]. ROP is a vasoproliferative disorder of the retina that can have adverse visual consequences, with blindness being the most severe.

The International Classification of Retinopathy of Prematurity (ICROP) provided a harmonized clinical classification for ROP [4]. The benefit of an ablative treatment to reduce the risk of retinal detachment was demonstrated for “threshold” ROP by the CRYO-ROP study. The ETROP (Early Treatment of Retinopathy of Prematurity) study introduced the definition of “pre-threshold” type 1 ROP and “pre-threshold” type 2 ROP [5,6]. Today, treatment guidelines are based on ETROP study recommendations. Treatment is indicated in "pre-threshold" type 1 ROP, while "pre-threshold" type 2 ROP requires close follow-up without treatment. Recently, the ICROP 3rd edition (ICROP3) study brought some updates to the ROP nomenclature: the definition of a posterior zone II region that begins at the margin between zone I and zone II, a definition of aggressive ROP that can occur in larger preterm infants and beyond the posterior retina, and precise definitions of regression and reactivation [7].

The ETROP classification introduced the definition of pre-threshold ROP in 2003 (Figure 1) [5,6]. Pre-threshold type 1 ROP includes: Zone I: any ROP stage with “plus” disease;Zone I: ROP stage 3 without “plus” disease;Zone II: ROP stage 2/3 with “plus” disease.

Meanwhile, pre-threshold type 2 ROP includes: Zone I: ROP stage 1/2 without “plus” disease;Zone II: ROP stage 3 without “plus” disease.

The ETROP study demonstrated that pre-threshold type 1 ROP treated earlier than in CRYO-ROP resulted in less unfavorable retinal structural outcomes at 6 years (15.2% of structural outcomes were unfavorable if treated at threshold in CRYO-ROP vs. 8.9% of eyes treated at pre-threshold in ETROP), thus expanding the treatment recommendations to pre-threshold type 1 ROP [6]. 

Treatment of pre-threshold type 1 ROP includes laser photoablation and intravitreal injection (IVI) of anti-VEGF [8,9]. Even today, laser photoablation is mostly used, due to its definitive character. Anti-VEGF IVIs are used to treat posterior ROP forms, and may be used for pre-threshold type 1 ROP in zone I (with expert consensus) [9,10]. Following the ETROP guidelines, pre-threshold type 2 or milder ROP is not eligible to treatment. ROP stages 4 and 5 both represent severe disease with a partial or total retinal detachment, requiring surgical treatment.

The Bevacizumab Eliminates the Angiogenic Threat for Retinopathy of Prematurity (BEAT-ROP) study has shown the superiority of bevacizumab intravitreal injection (IVI) compared to laser photoablation for stage 3 ROP in zone I with plus disease [8]. The most recent randomized trial—the RAINBOW multicenter trial, which used ranibizumab IVI—included ROP stage 3 in zone II with plus disease in its treatment recommendations [9]. Despite the absence of international guidelines, validated indications for IVI—suggested by expert consensus and a recent study—include aggressive ROP and pre-threshold type 1 in zone I [10]. In aggressive ROP, the vascularization ends in zone I or the posterior area of zone II, and is associated with “plus” disease. Aggressive ROP can progress rapidly.

Since 2003, the epidemiology of premature children has changed, with an increase in the survival of severely premature children. As a result, indications for treatment outside the ETROP guidelines, based on physicians’ clinical experience and collegial discussions, have been slowly increasing [11,12,13].

The purpose of this study was to determine the prevalence and characteristics of premature children treated for ROP less severe than pre-threshold type 1, and to describe the pathway to clinical decisions to use treatments outside the recommended guidelines. 

## 2. Materials and Methods

### 2.1. Study Design

In this retrospective monocentric observational cohort, we included all premature infants who received treatment for ROP at the Rothschild Foundation Hospital (RFH) between January 2016 and January 2020. Preterm infants receiving surgery as a first-line treatment (ROP stage 4 or 5) were excluded. This study was conducted in accordance with the Declaration of Helsinki and current French legislation, and with the approval of the local ethics committee.

### 2.2. Clinical Assessment

The following data were collected: gestational age (GA), birth weight (BW), sex, ETROP stage, ICROP classification, corrected age at the time of treatment, type of the first treatment (laser ablation or bevacizumab IVI), number of laser impacts if laser ablation was performed, screening method—wide-angle imaging system (RetCam^®^, Clarity Medical Systems, Pleasanton, CA, USA) or indirect ophthalmoscopy—and distance in kilometers between home and RFH (itinerary recommended by Google Maps^®^). 

According to the ICROP3 classification, we defined the “pre-plus” stage as the presence of abnormal vascular dilation and/or tortuosity insufficient for plus disease [7]. Pre-plus disease is an independent prognostic factor strongly associated with progression to severe ROP, requiring photoablative treatment [14]. 

Treatment indications were decided during a physical consultation, or following tele-interpretation of retinography images obtained using a wide-angle imaging system.

Treatment was performed under general anesthesia, and all infants were dilated using tropicamide 0.5% (Mydriaticum 0.5%, Laboratoires Théa, Clermont-Ferrand, France). During the exam under general anesthesia, the ETROP stage and the ICROP classification were reassessed before treatment using a QuadrAspheric^®^ (Volk Optical Inc., Mentor, OH, USA) pediatric lens with indentation completed via the acquisition of wide-angle images. Choice of treatment—laser or IVI—was left to the pediatric ophthalmologist’s discretion. Retinal photoablative treatment was performed using a frequency-doubled Nd–YAGr laser mounted on the microscope and a QuadrAspheric^®^ pediatric lens. In cases of anti-VEGF IVI, we injected 0.025 mL of bevacizumab 2 mm from the limbus. All infants received a follow-up examination one week after treatment. 

The main outcome of our study was the proportion of infants treated bilaterally outside the recommended guidelines. To ascertain this, our cohort was divided into two groups: Group 1 included infants treated outside the ETROP guidelines for both eyes for ROP less severe than pre-threshold type 1. Group 2 included infants who received treatment for at least one eye within the guidelines for a pre-threshold type 1 ROP. The clinical pathways resulting in treatment outside the guidelines for both eyes were analyzed. Treatment was decided based on literature review and current practices in our department, and included persistent stage 2 or 3 ROP that showed no evidence of regression after 41 weeks of corrected gestational age, pre-plus disease stage, difficulties in disease staging, pre-threshold type 2 ROP with plus disease, logistical difficulties, and complicated follow-up [12]. Signs of activity in persistent ROP were defined as the persistence of vascular tortuosity or the persistence of peripheral ischemia, confirmed by fluorescein angiography, beyond a vascularized or non-vascularized ridge. “Logistical difficulties” refer to preterm infants hospitalized in neonatal units hundreds of miles away from RFH, with no fundus examination possible on site, resulting in major difficulties in following these children. The “difficulties in disease staging” indication group concerned high-risk children screened using wide-angle imaging or indirect ophthalmoscopy on conscious children with suspicious but inconclusive preoperative exam results due to uncompliant children or low-quality image definition. Under general anesthesia, exams were easier to perform. Children in such cases were treated outside the guidelines because they were considered at high risk of progression, and so as to avoid a second general anesthesia. Furthermore, in our study design, the criterion “pre-threshold type 1 ROP in the fellow eye” was not considered as an off-label indication—since children were treated within the ETROP guidelines for at least one eye—and was included in group-2.

### 2.3. Statistical Analysis

Statistical analysis was carried out using MedCalc^®^ version 19.3 (MedCalc Software Ltd, Ostend, Belgium). Descriptive statistics were reported as the mean ± standard deviation (SD) and at a confidence interval of 95% (CI 95%) for continuous variables, and as percentages for categorical variables. The chi-squared test was used to compare the two groups for categorical variables, while ANOVA or Student’s *t*-test was used for quantitative variables. A *p*-value of less than 0.05 was considered to be statistically significant. 

## 3. Results

Of the 116 children treated during the inclusion period, 2 were excluded because they had one eye with stage 4/5 ROP that required surgical treatment. A total of 114 children were included, of whom 92 (80.7%) were treated by laser ablation and 22 (19.3%) by anti-VEGF IVI (bevacizumab). A total of 32 children (28.1%) were treated outside the ETROP guidelines for both eyes, and were included in group 1. Group 1 included 42 eyes (66.7%) presenting pre-threshold type 2 ROP and 21 eyes (33.3%) presenting at milder stages. A total of 82 (72.4%) children received treatment within the guidelines for at least one eye, and were included in group 2. Group 2 included 10 children treated within the guidelines for pre-threshold type 1 ROP in only one eye, and had the fellow eye treated despite an ROP milder than pre-threshold type 1 (9 eyes pre-threshold type 2 ROP and 1 milder stage). There were no significant differences between the two groups regarding the nature of the treatment (*p* = 0.98). Children in group 1 had an average GA of 25.3 ± 1.4 weeks and an average birth weight of 734 ± 174 grams. There were no significant differences in the baseline characteristics between the two groups (Table 1). The corrected age at the time of treatment was higher in group 1 (40.6 ± 7.0 weeks vs. 37.4 ± 3.6 weeks; *p* = 0.002). 

We identified five different clinical indications for treatment outside the recommended guidelines for both eyes: persistent stage 2 or 3 ROP that showed no evidence of regression after 41 weeks of corrected gestational age (11 children; 34.4%), pre-plus stage (11 children; 34.4%), difficulties in disease staging (7 children; 21.9%), type 2 ROP with plus disease (2 children; 6.2%), and logistical difficulties (1 child; 3.1 %) (Table 2). 

Children in group 2 had more laser impacts than those in group 1 (2007 ± 171 impacts vs. 1460 ± 207 impacts; *p* = 0.03). There was no significant difference in the distance between home and RFH between the two groups (*p* = 0.43). In this cohort, 87 children (76.3%) were screened using wide-angle imaging, while 27 children (23.7%) were screened using indirect ophthalmoscopy; there were no differences in the screening methods used between the two groups (*p* = 0.97). On the day of treatment, we noted a staging concordance with preoperative indirect ophthalmoscopy staging for only 10 children (37.0%). 

We analyzed the characteristics of the infants in group 1, stratified by clinical indications (Table 3). Infants treated for “non-resolved” ROP were older than the other children when treated (47.5 WG ± 7.7 vs. 40.6 WG ± 7.0; *p* = < 0.01). We noticed that children treated for “difficulty in disease staging” were younger than other subgroups (24.7 WG ± 0.6 vs. 25.3 WG ± 1.3; *p* = 0.03).

## 4. Discussion

In this retrospective study, we reviewed all infants with ROP who received treatment by laser ablation or IVI, and found that 32 children (28.1%) had ROP less severe than pre-threshold type 1 in both eyes. We chose to present results on children and not on eyes, because we consider that the pathway to clinical decision leading to treatment was based on the child and not the eye. If children needed treatment on one eye, general anesthesia was performed, and both eyes were treated in order to avoid the risk of a second general anesthesia. We now perform IVI with only sedation, but at the end of the inclusion period of our study this was not current practice for the department. 

Our results show a higher rate than what was reported in the literature. In 2016, Gupta et al. [11] reported 9.5% of eyes treated outside the guidelines in a multicenter American study, while Liu et al. [12], in their secondary analysis of data collected as part of the Postnatal Growth and ROP (G-ROP) study, reported a frequency of 12.5%; this difference could be due to several factors, such as a younger cohort or a more lenient treatment protocol. In France, ROP follow-up—especially after discharge of the child from a neonatal intensive care unit—is a logistical and organizational challenge because of the scarcity of pediatric ophthalmologists. This point leads experts to treat persistent and active ROP earlier out of concern for missing follow-up examinations, and because of the difficulties of fundus examination in older infants [15]. Furthermore, 27 children were screened using indirect ophthalmoscopy, of whom only 10 had a concordant staging between screening and the exam under general anesthesia; this may have led to more treatment outside indications.

The mean gestational age of children treated outside the recommended guidelines was 25.3 ± 1.4 weeks, similar to the results reported by Gupta et al. (25.3 weeks) [11]. Mean age at treatment was also similar to that in the reported literature: 40.6 weeks in our study, 41.5 weeks in the study of Gupta et al., and 38 weeks in the study of Liu et al. [12]. In medical ROP literature, many underlying reasons for off-label treatment have been reported. In our study, we retained these five indications outside the ETROP guidelines due to their frequency in ROP studies and their clinical relevance [11,12,13]. As mentioned above, we did not consider pre-threshold type 1 ROP in the fellow eye as an indication outside the guidelines, but we included these children in the ETROP-guidelines-treated group. In fact, the pathway to the clinical decision to treat was based not on the diagnosis of ROP milder than pre-threshold type 1, but on the presence of pre-threshold type 1 ROP in the fellow eye; in Liu et al.’s study, this represented the most common indication for off-label treatment, with 43% of eyes [12]; in our study, it was a minor phenomenon, as 32 children were treated for off-label indications, while only 10 children had treatment outside indications with a pre-threshold type 1 ROP in the fellow eye. We believe that the pathway leading to treatment of the fellow eye for pre-threshold type 1 ROP is the fear of the risk related to additional general anesthesia in these infants, which is different from the pathway leading to treatment of infants outside the indications [16].

With respect to indications outside the recommended guidelines, approximately 34% of children were treated because of unresolved ROP (Figure 2). These premature infants had an active ROP at a corrected age ≥ 41 weeks, with a mean age of 47.5 ± 7.5 weeks. Similarly, 30.8% of children in the study of Gupta et al. were treated because of persistent ROP at an advanced postmenstrual age [11]. Several studies showed that treating persistent stage 3 ROP after 41 weeks could decrease the risk of anatomical complications, such as anteroposterior traction or temporal vessel straightening [11]. The new ICROP3 study defined the regression of the disease well, but did not highlight the case of persistent stage 3 ROP [7]. 

Pre-plus disease represented ~34% of children treated outside the guidelines. The “pre-plus stage” is an independent prognostic factor and a marker of future disease aggravation [12,14]. Furthermore, the boundary between pre-plus and plus stage is sometimes blurred, and classification usually depends of the subjectivity and the experience of the physician. Some ROP experts tend to over-stage pre-plus disease in order to meet the ETROP criteria for treatment. In Japan, for example, the classification of ROP is slightly different, and many ROP stage 2 cases are considered stage 3 “early”, which results in the widening of treatment indications [4]. Based on our study and previous studies, as well as current practice, it is interesting to consider the inclusion of the “pre-plus” stage 3 zone II in the treatment indications. 

Approximately 22% of children were over-treated due to "difficulty in disease staging". Whether by wide-angle imaging or by indirect ophthalmoscopy, sometimes the initial staging established at the preoperative exam was not qualitative and did not enable us to obtain a concordance with the definitive and real ROP staging performed under general anesthesia at the time of treatment. In these cases, treatment was applied because of the benefit of treating these pre-threshold type 2 ROP cases rather than performing additional general anesthesia in case of an evolution to pre-threshold type 1 ROP. Moreover, we found a younger gestational age at birth for these children than for other subgroups. These preterm infants may be more difficult to examine in a perfect manner.

Most of our patients received laser ablation as a first-line treatment (80.7%). Only 19.3% received bevacizumab IVI. As with other studies, laser photoablation is still the most common treatment for ROP disease. This reflects real practice and surgeon habits. Bevacizumab IVI indications are limited to aggressive posterior ROP or zone I or posterior zone II disease. In our study, when both laser ablation and IVI treatment were valid, ROP experts tended to favor laser treatment. Due to the lack of long-term data on the effects of anti-VEGF agents on children [17], ROP experts may still be reluctant to consider this therapeutic option when laser ablation is available. Additionally, the definitive nature of a well-performed laser treatment is another reason to prefer it. 

The strengths of this study are the large number of eyes included and the uniformity in the treatment indications, since all children were treated in a single tertiary center. The present study was retrospective, and treatment indications were dependent on the subjectivity and the experience of the evaluating physician. Despite being a limitation to our study, this clearly reflects our real-life practice. 

In conclusion, treatment of premature infants with ROP less severe than pre-threshold type 1 is common, and raises several questions. With the recent improvement of neonatal intensive care, the kinetics of ROP may have changed with the survival of more extreme premature children and the increase in active, unresolved ROP. Guidelines resulting from the ETROP study published in 2003 may not represent the actual epidemiology of ROP, and revision may be necessary [18]. Additional studies are required in order to evaluate the long-term impacts of treatments outside the recommended indications, and to establish revised treatment guidelines. 

## Figures and Tables

**Figure 1 jcm-11-00039-f001:**
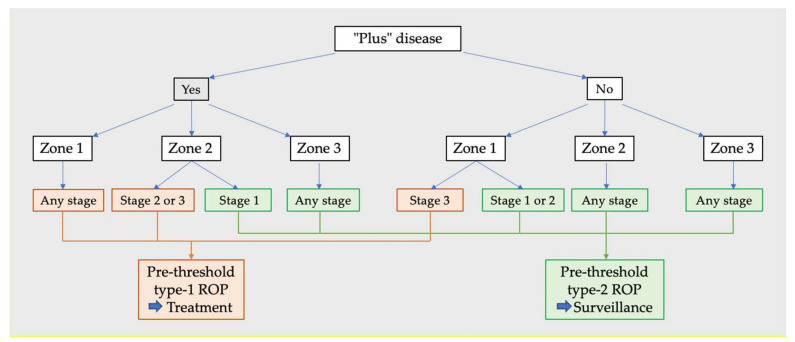
Pre-threshold type 1 and type 2 ROP (Retinopathy of Prematurity).

**Figure 2 jcm-11-00039-f002:**
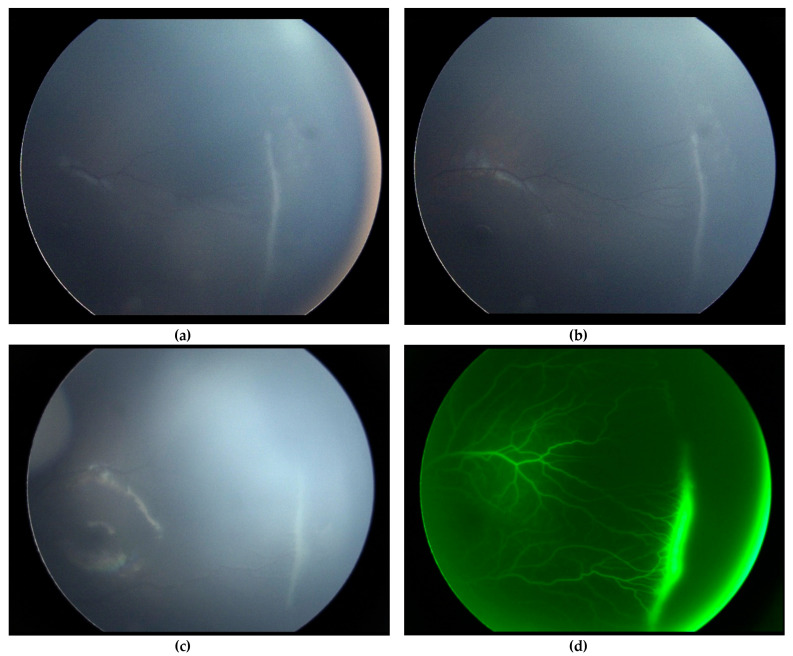
Example of “non-resolved” ROP at 40 weeks of amenorrhea-corrected age. Both upper retinography images (**a**,**b**) show “non-resolved” ROP at 40 weeks of amenorrhea-corrected age. We can visualize the ridge and the avascular retina beyond it. The limit of vascularization is located at posterior zone II. The ICROP3 stage of this eye is stage 3 posterior zone II, without plus or pre-plus disease. Both lower images (**c**,**d**) show the same eye two weeks later. We can see in this retinography the persistence of the ridge that had not regressed, with the non-vascularized retina beyond. The right image (**d**) is a fluorescein angiograph taken under general anesthesia on the day of the treatment. We can clearly see a fluorescein diffusion around the ridge. This baby received a photoablative treatment following this exam due to the “non-resolved” character of the ROP beyond 42 weeks of gestation.

**Table 1 jcm-11-00039-t001:** Characteristics of the study population.

Features	Study Population (Group 1 and Group 2)	Group 1 Children Who Received Treatment within the Guidelines for Both Eyes (*n* = 32)	Group 2 Children Treated within the Guidelines for at Least One Eye (*n* = 82)	*p* Value *
Birth weight ± SD	731.8 ± 174.2	733.7 ± 174.0	730.6 ± 175.5	0.94
Weeks of gestation ± SD	25.4 ± 1.4	25.3 ± 1.4	25.5 ± 1.3	0.93
Corrected age at time of treatment ± SD	38.4 ± 5.0	40.6 ± 7.0	37.4 ± 3.6	0.002
Gender (%)				0.95
Male	68 (60%)	21 (66%)	47 (57%)	
Method of recruitment Wide-angle imaging (%)	87 (76%)	23 (72%)	64 (78%)	0.97
Clinical fundus	27 (24%)	9 (28%)	18 (22%)	
Treatment				0.98
Laser ablation IVI of anti-VEGF	92 (81%) 22 (19%)	27 (84%) 5 (16%)	65 (81%) 17 (19%)	

ROP: Retinopathy of prematurity; IVI: intravitreal injection; SD: standard deviation; WG: weeks of gestation. * χ^2^ test performed to compare groups 1 and 2 with one another.

**Table 2 jcm-11-00039-t002:** Characteristics of off-label-treated eyes (group 1).

		Number of Children (%)	Number of Eyes (%)	Treatment (Number of Eyes)		ICROP Eye Classification				
						Pre-threshold Type 2 ROP				Milder than Type 2 ROP
				Laser	IVI	S3Z2PP	S3Z2-	S2Z1	S2Z1PP	
Indications for treatment	Non-resolved ROP	11 (34.4%)	22 (34.9%)	22		3	8			11
	Pre-plus-stage ROP	11 (34.4%)	21 (33.3%)	21		18	1			2
	Difficulty in disease staging	7 (21.9%)	14 (22.2%)	6	8		7	3	2	2
	Non-type-1 plus-stage ROP	2 (6.2%)	4 (6.4%)	4						4
	Logistic difficulties	1 (3.1%)	2 (3.2%)		2					2
Treatment	Argon Laser		53 (84.1%)			21	12	1		19
	Bevacizumab IVI		10 (15.9%)				4	2	2	2
Total		32	63			21	16	3	2	21

ETROP; Early Treatment of Retinopathy of Prematurity Study; ICROP; International Classification of Retinopathy of Prematurity; IVI; intravitreal injection; S3Z2PP; stage 3 zone II pre-plus; S3Z2-; stage 3 zone II no longer or pre-plus; S2Z1-; stage 2 zone I no longer or no longer pre-existing; S2Z2PP; stage 2 zone 2 pre-existing; S2Z2-; stage 2 zone 2 no longer or no longer pre-existing; S2Z3PP/-; stage 2 zone 3 no longer; S3Z3+; stage 3 zone 3 pre-existing; S3Z3PP/; stage 3 zone 3 pre-existing; S1; stage 1 any zone.

**Table 3 jcm-11-00039-t003:** Group 1 indications characteristics.

			Characteristic Items ± SD					
		Number of Children	Birth Weight (g)	*p*	Weeks of Gestation (WG)	*p*	Corrected Ageat Treatment	*p*
All		32	733.7 ± 174.5		25.3 ± 1.3		40.6 ± 7.0	
Indications	Non-resolved	11	743.2 ± 140	0.81	25.3 ± 1.5	0.99	47.5 ± 7.7	**<0.01**
	Pre-plus stage	11	668.6 ± 87.3	0.06	25.2 ± 0.8	0.47	36.9 ± 2.1	**<0.01**
	Difficulty in disease staging	7	718.3 ± 138.8	0.78	24.7 ± 0.6	**0.03**	35.4 ± 1.4	**0.03**
	Non type 1 ROP stage plus	2	1152.5 ± 350	0.32	27.9 ± 2.9	0.41	37.8 ± 1.5	0.12
	Logistic difficulties	1	600.0		26.9 SA		37.7 SA	

SD: standard deviation; Student’s *t*-test was used to compare different indication groups to the rest of the patients included in group 1. Significant *p* values are marked with bold.

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
