# Peer review of "Treatment outside the Recommended Guidelines for Retinopathy of Prematurity (ROP): Prevalence, Characteristics, and Issues"

_jcm, 2021, doi:10.3390/jcm11010039_

Round 1
Reviewer 1 Report
The authors appropriately highlighted the aims, significance and novelty of their work. The methods and statistical analyses used seem to be appropriate. The reliability and validity of the results seem to be rigid. The conclusions are supported by the data presented.
In this study, the authors evaluated the prevalence and characteristics of retinopathy of prematurity (ROP) treated outside of recommended guidelines.
Defining non-guideline lesions is challenging, but the items were well considered and selected.
The findings of this study, which indicate the need for treatment of lesions outside of the guidelines, are significant in that they may influence the revision of the current guidelines.
Author Response
Point 1: In this study, the authors evaluated the prevalence and characteristics of retinopathy of prematurity (ROP) treated outside of recommended guidelines.
Defining non-guideline lesions is challenging, but the items were well considered and selected.
The findings of this study, which indicate the need for treatment of lesions outside of the guidelines, are significant in that they may influence the revision of the current guidelines.
Response 1: Thank you for your review.
Best regards
Reviewer 2 Report
The object of this article is interesting and deserves being contemplated and thoroughly studied. Some points need to be more explicit for non ped specialists who wish to learn about the subject. We congratulate the authors for picking up this specialized yet important subject for public health.
There are a couple of concerns that need to be addressed.
Abstract:
- Abstract is very important to readers, please expand further on the term “logistic difficulties”
- The last sentence of the abstract is not clear.
Introduction:
As a general remark, it is confusing to talk about both threshold and pre-threshold ROP. Please review and edit figure 1 accordingly. To be more specific change type 1 ROP for pre-threshold type 1 ROP throughout the article.
As a general remark, it would be appreciated to add a table presenting the currently accepted treatment indications for ROP.
Methods:
Please be clearer on the “on-label” indication of your group 2. Did you include children with either one or two eyes treated inside the ETROP guidelines? If so, please review and modify the manuscript accordingly, whenever applicable.
The difficulties in disease staging indication group concerned children screened using wide-angle imaging or indirect ophthalmoscopy with an inconclusive preoperative exam to stage correctly the ROP. Under general anesthesia, exam was easier performed. Children were in that case treated outside guidelines because they were considered at high risk of progression and to avoid a second general anesthesia. » :
Please expand on the difficulties in disease staging (exam on awake children? Image definition). Please edit this paragraph to make it more understandable. If I understood well, children in this group were overtreated to avoid other exams under general anesthesia for follow-up?
Please modify : “exam was easier to perform”
Please precise what “logistical difficulties” refer to.
Results:
Again, after the methods section, state clearly that group 1 includes children treated outside the guidelines and group 2 those treated according to the guidelines in at least one eye.
Table 1 you may add a column to present the entire population (group 1 + group 2)
The objective of the study was to present and understand the reasons for indicating an off-label treatment in ROP. I was wondering why you choose to present results on children and not on eyes? You may discuss this at the beginning of your discussion section.
Please explain how you collected the reasons for off label therapy.
It is confusing why table 3 refers to eyes and not individuals. Please delete or choose only one way to present your results.
What type of testsa did you perform in Table 4. Please add at the bottom of the table.
Table 1: delete WG from the columns. Man and woman should be replaced by either male or female (and not both since they are exclusionary). Vitreous and not vitrous.
Table 2 is unfortunately not readable; the text is shifted and columns do not correspond to rows. Please provide a new version of the table.
Author Response
ABSTRACT
Point 1: Abstract is very important to readers, please expand further on the term “logistic difficulties”
The last sentence of the abstract is not clear. 

Response 1: We thank the reviewer for his remark. We detailed the term “logictic difficulties” in the abstract, and we expand further on it in methods section. We have also reworded the last sentence of the abstract.
INTRODUCTION
Point 2: As a general remark, it is confusing to talk about both threshold and pre-threshold ROP. Please review and edit figure 1 accordingly. To be more specific change type 1 ROP for pre-threshold type 1 ROP throughout the article.
Response 2: We changed the terms from “type 1 ROP” to “pre threshold type1 ROP” throughout the article. We also review figure 1 accordingly.
Point 3: As a general remark, it would be appreciated to add a table presenting the currently accepted treatment indications for ROP.
Response 3: We added in the boxed text precisions about currently accepted treatment indications for ROP
METHODS
Point 4: Please be clearer on the “on-label” indication of your group 2. Did you include children with either one or two eyes treated inside the ETROP guidelines? If so, please review and modify the manuscript accordingly, whenever applicable.
Response 4: We brought precisions about “on-label” indication of group 2. Throughout method section, results and discussion, we clearly defined which children were included in this group. Group-1 included infants treated outside ETROP guidelines both eyes for ROP less severe than pre-threshold type 1. Group-2 included infants who received treatment with at least one eye inside ETROP guidelines for a pre-threshold type 1 ROP.
Point 5: Please expand on the difficulties in disease staging (exam on awake children? Image definition). Please edit this paragraph to make it more understandable. If I understood well, children in this group were overtreated to avoid other exams under general anaesthesia for follow-up?
Response 5: Yes, these children were overtreated to avoid new general anesthesia due high risk of evolution to a pre-threshold type 1 ROP. We explicit more clearly this concept in the reviewed article, and also we bring more details about difficulties to stage ROP disease: “The “difficulties in disease staging” indication group concerned high risk children screened using wide-angle imaging or indirect ophthalmoscopy on awake children with suspicious but inconclusive pre-operative exam due to uncompliant children or low-quality image definition”
RESULTS
Point 6: Table 1 you may add a column to present the entire population (group 1 + group 2). It is confusing why table 3 refers to eyes and not individuals. Please delete or choose only one way to present your results.
Response 6: As advised, we add a column on table 1 to present the entire population. Furthermore, we delete table 3 to avoid confusion.
Point 7: The objective of the study was to present and understand the reasons for indicating an off-label treatment in ROP. I was wondering why you choose to present results on children and not on eyes? You may discuss this at the beginning of your discussion section.
Response 7: We explain more clearly why we chose to present results on children and not on eyes.
“We chose to present results on children and not on eyes because we consider pathway to clinical decision leading to treatment was based upon child entity and not eye. If children need treatment on one eye, general anesthesia will be performed and both eyes will be treated to avoid the risk of a second general anesthesia”
Point 8: Please explain how you collected the reasons for off label therapy.
Response 8: We add in discussion section explanation about the collection of indications outside the guidelines.
“In medical ROP literature, many underlying reasons for off-label treatment were re-ported. In our study, we retained these five indications outside the ETROP guidelines due to their frequencies in ROP studies and their clinical relevance [11-13].”
Point 9: Table 2 is unfortunately not readable; the text is shifted and columns do not correspond to rows. Please provide a new version of the table
Response 9: Sorry for the inconvenience, we provide a new Table 2.
Point 10: What type of testsa did you perform in Table 4. Please add at the bottom of the table.
Response 10: We did a Student’s T-test, each sub-category versus the rest of patients of group 1. We added information at the bottom of the table.